# The OMG dataset: An Open MetaGenomic corpus for mixed-modality genomic language modeling

**Andre Cornman**[*,1]  **Jacob West-Roberts**[1]  **Antonio Pedro Camargo**[2]
**Simon Roux**[2]  **Martin Beracochea**[3]  **Milot Mirdita**[4]  **Sergey Ovchinnikov**[5]
**Yunha Hwang**[*,1]

[1]Tatta Bio, USA
[2]DOE Joint Genome Institute, Lawrence Berkeley National Laboratory, Berkeley, CA, USA
[3]European Molecular Biology Laboratory, European Bioinformatics Institute (EMBL-EBI), Wellcome Genome Campus, Hinxton, Cambridge, UK
[4]School of Biological Sciences, Seoul National University, Seoul, Republic of Korea
[5]Department of Biology, Massachusetts Institute of Technology, Cambridge, MA, USA
[*]Correspondence: `{yunha,andre}@tatta.bio`

## Abstract

Biological language model performance depends heavily on pretraining data quality, diversity, and size. While metagenomic datasets feature enormous biological diversity, their utilization as pretraining data has been limited due to challenges in data accessibility, quality filtering and deduplication. Here, we present the Open MetaGenomic (OMG) corpus, a genomic pretraining dataset totalling 3.1T base pairs and 3.3B protein coding sequences, obtained by combining two largest metagenomic dataset repositories (JGI's IMG and EMBL's MGnify). We first document the composition of the dataset and describe the quality filtering steps taken to remove poor quality data. We make the OMG corpus available as a mixed-modality genomic sequence dataset that represents multi-gene encoding genomic sequences with translated amino acids for protein coding sequences, and nucleic acids for intergenic sequences. We train the first mixed-modality genomic language model (gLM2) that leverages genomic context information to learn robust functional representations, as well as coevolutionary signals in protein-protein interfaces and genomic regulatory syntax. Furthermore, we show that deduplication in embedding space can be used to balance the corpus, demonstrating improved performance on downstream tasks. The OMG dataset is publicly hosted on the Hugging Face Hub at `https://huggingface.co/datasets/tattabio/OMG` and gLM2 is available at `https://huggingface.co/tattabio/gLM2_650M`.

## 1 Introduction

Biological language models present an effective avenue for leveraging large amounts of unstructured sequence data and learn functionally meaningful representations. Similar to natural language processing (NLP) models (Touvron et al., 2023; Dodge et al., 2021), the quality and diversity of pretraining data dictate the behavior and performance of biological language models (Ding & Steinhardt, 2024). To date, the most widely used datasets for biological language models (Hayes et al., 2024; Lin et al., 2023; Madani et al., 2023; Nguyen et al., 2024) are derived from curated data repositories such as UniProt (UniProt Consortium, 2019), UniRef (Suzek et al., 2007) and GTDB (Parks et al., 2022). However, biological sequence diversity is immense and the above-mentioned data repositories cover only a small fraction of the full sequence diversity found in nature. In order for biological

language models to improve, the size and diversity of pretraining data must also scale with the size of the model.

Metagenomic sequences are partial genomic sequences derived from direct sequencing of environmental (e.g. soil, ocean) or biological samples (e.g. human skin, gut). Because metagenomic sequencing circumvents the need for cultivation and isolation of biological organisms, metagenomes typically feature sequences derived from uncultivated and novel microorganisms (Tyson et al., 2004). These microbial genomes encode high levels of molecular diversity and span previously unexplored branches of the tree of life (Hug et al., 2016). Metagenomic datasets are unstructured by nature and a large fraction of the data is not included in curated databases due to poor functional interpretability of these sequences. To date, metagenomic sequences have not been fully utilized in biological language models due to following limitations:

1. **Metagenomic sequences are not readily downloadable in a single archive.** To date, the download of raw contigs (assembled genomic segments) from the two main public repositories, Joint Genome Institute (JGI)'s IMG (Markowitz et al., 2012) and European Molecular Biological Laboratory (EMBL)'s MGnify (Richardson et al., 2023), requires a large number of database queries and/or rate-limited web API calls, as well as ad hoc approaches to robustly aggregate the results of these queries into a single dataset.

2. **Metagenomic sequences require extensive pre-processing.** Raw metagenomically assembled contigs first undergo gene calling in order to identify protein coding sequences and extract translated sequences. Additional quality filtering is critical, as many metagenomes include poor or mis-assembled contigs.

3. **Metagenomic sequences are difficult to deduplicate and balance.** Like most biological sequence datasets, metagenomes feature sampling biases (e.g. over-representation of human gut microbiomes). Additionally, due to the lack of centralized databases for metagenomes, submissions of identical metagenomes to different repositories result in duplicates. Unlike protein databases that can be deduplicated and balanced using computationally efficient clustering algorithms (e.g. MMseqs2 (Steinegger & Söding, 2017)), clustering of a large dataset comprising genomic sequences of arbitrary region and length is computationally costly. Furthermore, while curated genomic databases (e.g. GTDB (Parks et al., 2022) or BV-BRC (Olson et al., 2023)) can be balanced with taxonomic labels, metagenomic sequences rarely have taxonomic assignment, and ad-hoc assignment (e.g. Kraken (Wood & Salzberg, 2014)) is computationally expensive and not always reliable.

Here, we document the collection and preprocessing steps of the OpenMetaGenome (OMG) corpus. We then train the first mixed-modality genomic language model (gLM2) trained on OMG, which leverages genomic context information to learn contextualized functional representations of genomic elements. By training on mixed-modality data, gLM2 can perform both protein and DNA downstream tasks, and outperforms ESM2 (Lin et al., 2023) on most protein tasks. Additionally, training on multi-protein contexts enables gLM2 to predict protein-protein interfaces through co-evolutionary signal. Finally, we show that embedding-based deduplication of the OMG dataset leads to improved functional representations, especially for underrepresented sequences.

## 2 Related Works

**Pretraining corpora preprocessing in NLP.** A number of previous studies have developed methods to improve the diversity and quality of pretraining corpora in NLP. For instance, raw snapshots of Common Crawl (collection of webtext crawls) contain undesirable data (e.g. hate speech, placeholder text). Studies have demonstrated that careful deduplication and rule-based filtering of Common Crawl (Dodge et al., 2021) improves overall model performance (Penedo et al., 2024). More recently, efforts have been made to prune and balance pre-training data in semantic embedding space to achieve increased training efficiency (Sorscher et al., 2022; Tirumala et al., 2023; Abbas et al., 2023). Dataset preprocessing

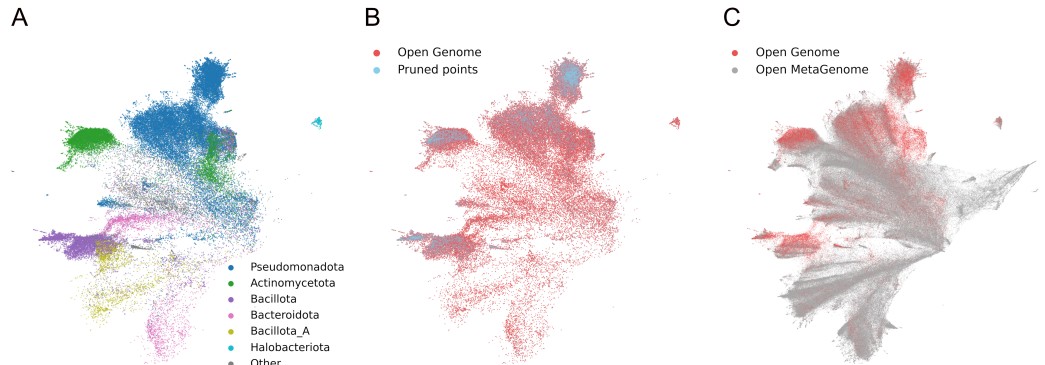

Figure 1: **(A)** UMAP visualization of the OG dataset examples, colored by taxonomic phylum, using embeddings from the 150M parameter gLM2 model. Distinct clusters form for different phyla in embedding space. **(B)** Semantic deduplication of the OG dataset, with pruned points highlighted in blue. Deduplication primarily removes samples from dense clusters corresponding to overrepresented phyla. We visualize the semantic deduplication on OG dataset to highlight taxonomic phyla most heavily pruned, and apply the same pruning process to the OMG dataset for model training. **(C)** Comparison of the OG and OMG datasets using a random 0.1% subset of each. Notably, the metagenomic data (OMG) exhibits higher diversity.

presents an important opportunity to minimize training resources, given the power-law nature of LLM scaling (i.e. exponentially increasing compute requirement for diminishing returns in performance improvement) (Hestness et al., 2017; Kaplan et al., 2020).

**Biological sequence language models and their training datasets.** Biological sequence language models are self-supervised models trained on discrete protein sequences or genomic segments. Protein language models (pLMs) (Lin et al., 2023; Madani et al., 2023; Elnaggar et al., 2022) are typically trained on high quality and curated publicly available datasets such as UniRef (Suzek et al., 2007). UniRef is convenient for pLM training because it has been deduplicated using sequence similarity-based clustering (i.e. UniRef50 is deduplicated using 50% sequence identity). Previous efforts to increase the diversity of the pretraining data includes cluster-balanced sampling (e.g. UniRef50/D for ESM models (Rives et al., 2021) and sequence identity-based clustering of compiled protein databases beyond curated databases (e.g. BFD (Steinegger et al., 2019; Elnaggar et al., 2022)). Genomic language models (gLMs) are trained on genomic sequences chunked at predefined length thresholds. Diversification efforts for genomic datasets include pretraining on MGnify's metagenomic contigs (Hwang et al., 2024) and balancing efforts in genomic pretraining datasets include taxonomy-aware sampling (Dalla-Torre et al., 2023; Nguyen et al., 2024) of curated genomic databases such as RefSeq (Pruitt et al., 2014), IMG/VR (Camargo et al., 2022), IMG/PR (Camargo et al., 2024) and GTDB (Parks et al., 2022).

**Metagenomic datasets.** In this study, we define metagenomic datasets as collections of genomic contigs (contiguous genomic segments) computationally assembled from either short-read or long-read raw sequence libraries. Typically, metagenomic datasets are sequenced from mixed community samples, which consist of multiple species, ranging from hundreds to thousands of distinct species (Bahram et al., 2021). Complete genomes are rarely obtained from metagenomic assemblies. Therefore, metagenomic assemblies require extensive taxonomic profiling (Parks et al., 2021) and partial genome reconstruction through contig clustering (i.e. binning). Because metagenomes are sequenced from diverse environments without the need for cultivation, their sequences feature the highest level of molecular diversity amongst publicly available sequence datasets (Pavlopoulos et al., 2023). Metagenomic datasets also vary in quality depending on sequencing depth and sample type, where low quality metagenomes feature computational assembly errors, short contig lengths, and truncated protein sequences (Mende et al., 2012; Lai et al., 2022). Furthermore, while most

metagenomic datasets are predominantly analyzed with a focus on microbial (archaea, bacteria, viruses) communities, eukaryotic genomic material can comprise a substantial portion of the raw library (West et al., 2018). Many standard metagenomic post-processing steps (e.g. gene calling) fail on eukaryotic sequences, resulting in poor quality protein sequence predictions. Critically, quality filtering and dataset deduplication of metagenomes require domain-specific knowledge, yet there is little documentation of preprocessing steps needed to make these datasets suitable for biological language model pretraining. While pretraining on metagenomic datasets allows models to leverage rich molecular diversity and genomic context, these models are most suitable for microbial genomes and may result in out-of-distribution effects on eukaryotic sequences.

## 3  THE OPEN METAGENOME CORPUS

Here, we document the construction of the OMG corpus. The OMG is a 3.1T base pair (bp) pretraining dataset comprising EMBL's MGnify database[1] and JGI's IMG database[2]. We utilize the gene predictions conducted by the databases; the gene calling protocols for IMG and MGnify are detailed in Huntemann et al. (2016) and Richardson et al. (2023) respectively. The combined dataset is pre-processed into a mixed-modality dataset upon sequential element-by-element quality-filtering (described in Section 3.1) . The mixed-modality dataset of Open Metagenomes is made available as the OMG dataset (Fig. 1) containing 3.3 billion protein coding sequences (CDS) (Tab. 1). We also make available a  10x smaller subset of OMG that only consists of prokaryotic and viral genomes from INSDC[3] as the Open Genome mixed-modality dataset OG (Fig. 1, Appendix B). Finally, we make available a protein-only dataset OMG_prot50, consisting of protein sequences derived from the OMG dataset, clustered at 50% sequence identity (Appendix E). OMG_prot50 contains 207M representative sequences from clusters with at least two members, representing >3-fold increase in sequence diversity compared to UniRef50 (Suzek et al., 2007). All three datasets are available for download from the Hugging Face Hub, and all dataset processing scripts are available at `https://github.com/TattaBio/OMG`. As more metagenomic data becomes available, we plan on regular updated releases of the corpus in the future.

Table 1: **Statistics for the datasets made available in this study.** CDS: Coding sequences, IGS: Intergenic sequences. For reference, UniRef50 consists of 66M proteins.

| | # CDS | # IGS | Total (bps) | # Contig | Size (TB) | Description |
|---|---|---|---|---|---|---|
| **OMG** | 3.3B | 2.8B | 3.1T | 271M | 1.25 | Mixed-modality genomic sequences with multiple protein coding genes (represented in AAs) interleaved with intergenic sequences (represented in NAs). |
| **OG** | 0.4B | 0.3B | 0.4T | 6.2M | 0.16 | Fraction IMG data consisting of prokaryotic genomes with taxonomic metadata. |
| **OMG_prot50** | 207M | – | – | – | 0.05 | Protein coding sequences clustered at 50% sequence identity, excluding singleton clusters. Clustering details in Appendix E |

---

[1]Snapshot date 2022-11-23 (excluding all embargoed/restricted metagenomic samples, see database statistics in Appendix A)

[2]Snapshot date 2023-08-27 (excluding all embargoed/restricted metagenomic samples and including IMG genomes dataset derived from NCBI.)

[3]`https://www.insdc.org`, retrieved from IMG/M, metadata available in Appendix P

## 3.1 Dataset preprocessing

**Multi-modal data processing.** Metagenomic contigs often encode multiple genes on either strand of the sequence. A genomic language model can be trained on raw nucleic acid sequences (e.g. Evo (Nguyen et al., 2024), Nucleotide Transformers (Dalla-Torre et al., 2023)) or by representing each genomic sequence as an order- and orientation-preserved list of translated coding sequences in amino acids (e.g. (Hwang et al., 2024)). For the former method, the context length needed to encode genomic sequences in nucleic acids can result in unfeasibly large compute requirements. Furthermore, a recent study comparing nucleic acid (NA) models against amino acid (AA) models on protein functional representations demonstrated that NA may not be the most efficient input format for learning translated protein functions (West-Roberts et al., 2024). The latter method, while benefiting from the compressed sequence length and more expressive AA sequences for proteins, does not leverage the information stored in intergenic regions. These intergenic regions contain important, yet, lesser characterized sequence patterns involved in transcription regulation and cellular function such as ncRNA, microRNA, promoters, and transcription factor binding sites. We developed a mixed-modality dataset that represents a genomic contig as a list of elements where an element is either a coding sequence (CDS) or an intergenic sequence (IGS) (see Fig. 2). CDS elements are represented in translated AA sequences and IGS elements are represented in NA sequences. We also store the strand information (+/-) of CDS elements and the order of all elements in the contig.

**Edge-element removal.** Metagenomic contigs are not complete genomic sequences, therefore, both edges of the sequences are more likely to contain gene-calling errors. In our pre-processing, we remove edge CDS elements to address miscalled open reading frames (ORFs) and fragmented protein sequences at the beginning and end of the metagenomic contigs (Steinegger & Salzberg, 2020). Specifically, if a scaffold starts/ends with an interrupted CDS, we remove that CDS element. If a scaffold starts/ends with a non-coding region, we remove the IGS element and the CDS adjacent to the IGS element. This filtering step removes ~1.4B genomic elements likely to be poor quality, partial sequences with high likelihood of assembly errors.

**Contig length-based filtering and preprocessing.** Assembly of shotgun metagenomic libraries results in many short contigs that are often low in quality. To limit the impact of the fragmented nature of metagenome assemblies, we first remove all metagenomic contigs that are shorter than 2kb from the raw databases. Secondly, we enrich the corpus with contigs that contain multiple genes by removing contigs that contain less than seven elements in total or less than three CDS elements. Only contigs that meet the length requirement are added to the dataset. In preprocessing these contigs into Hugging Face datasets (Lhoest et al., 2021), we found that extremely large contigs resulted in process hanging errors and inefficient storage. To address this issue, we chunk large contigs into 1000 elements. Appendix C visualizes the distribution of contig length, as well as CDS and IGS element lengths.

**Assembly quality (N/X-frequency) filtering.** Due to the computational nature of the metagenomic assembly, misassembled contigs comprise a nontrivial fraction of the data. The quality of the assembly differs significantly across samples, depending on the biological community composition, sample type, and sequencing depth (Vollmers et al., 2017; Lapidus & Korobeynikov, 2021). Notably, the quality of assembly may vary across the contig, where a section of the contig may contain assembly gaps due to shallow sequencing depth. One way to determine poorly assembled sequences is by identifying the fraction of Ns (gaps or ambiguous bases) in the raw DNA sequence (or Xs in the translated AA sequence). For OMG, we process each contig sequentially element-by-element, and if an element comprises >20% in invalid characters, we discard the element and start a new contig (Appendix. D). Importantly, only contigs that meet the length requirement above are added to the dataset. This sequential processing allows high quality regions of the contigs to be preserved, while low quality stretches are discarded.

**Element length-based filtering.** A nontrivial portion of the metagenome can be eukaryotic, however, most metagenomic gene-calling software tools are not optimized for eukaryotic

ORF prediction (Bruna et al., 2024). Additionally, metagenomes can contain sequences from organisms that utilize alternative genetic codes (Borges et al., 2022; Cook et al., 2024), which may not all be correctly predicted by common tools. A salient pattern observed for poor gene prediction is low coding density, (i.e. long stretches of IGS) or presence of very long CDS sequences. To identify these, we process each contig sequentially element-by-element and remove any CDS element >15,000 AAs or IGS element >4000 bps in length, and start a new contig. These thresholds are designed to exclude regions of questionable gene calls, such as long intergenic regions where no genes are predicted, and giant protein sequences, which are prone to assembly errors and require careful curation to verify (West-Roberts et al., 2023). This filtering step removes 2.5e-5% of CDS , and 1e-4% of IGS elements from OMG.

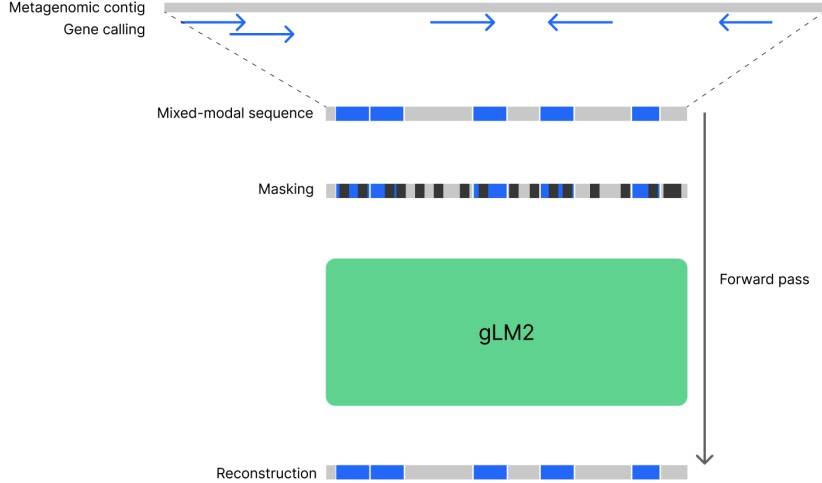

Figure 2: **Mixed-modality sequence processing and gLM2 masked language modeling.** A gene-called metagenomic contig is first preprocessed into a mixed-modality sequence consisting of CDS elements (blue) and IGS elements (grey). The mixed-modality sequence then undergoes masking at 30% and gLM2 is trained with a masked token reconstruction objective.

## 4 EXPERIMENTS

### 4.1 GLM2: A MIXED-MODALITY GENOMIC LANGUAGE MODEL

To showcase the efficacy of the OMG dataset for pretraining, we introduce gLM2: a mixed-modality genomic language model pretrained on OMG. gLM2 learns contextualized representations of genomic contigs, which are represented as sequences of CDS and IGS elements. In order to tokenize the mixed-modality sequence, CDS elements are tokenized using per-amino acid tokens, and IGS elements are tokenized using per-nucleotide tokens. To distinguish strand orientation for CDS elements, we introduce two special tokens: `<+>` and `<->`, which are prepended to each genomic element to indicate the positive and negative strands, respectively. gLM2 is trained using the masked language modeling objective, where 30% of both CDS and IGS tokens are masked. Cross-entropy loss is applied only on the masked tokens. gLM2 is trained at two scales: 150M and 650M parameters. Both models are trained on the semantically deduplicated OMG dataset (Section 4.2) for 600k steps. We train gLM2 using a context window of 4096 tokens to allow for multiple (9.7 ± 3.3) CDS and IGS elements to appear in each example. For model architecture and training hyperparameters, refer to Appendix F.

We benchmark gLM2 on the Diverse Genomic Embedding Benchmark (DGEB) (West-Roberts et al., 2024). DGEB is a comprehensive benchmark that evaluates model represen-

tations across diverse taxa and 18 tasks representing multiple axes of biological function, such as evolutionary distance similarity, remote homology prediction, enzyme classification, and retrieval sensitivity.

## 4.2  OMG CORPUS BALANCING WITH GENOMIC SEMANTIC DEDUPLICATION

Biological datasets exhibit significant biases that can influence the performance and generalizability of trained models (Ding & Steinhardt, 2024; West-Roberts et al., 2024). Unlike protein databases, where short sequence lengths allow for clustering-based deduplication, (meta)genomic sequences have highly variable lengths (Appendix C), making sequence-based clustering challenging. To address this challenge, we perform deduplication in embedding space by pruning examples with small cosine distance, following Semantic Deduplication (SemDeDup) (Abbas et al., 2023). SemDeDup previously showed efficacy in removing semantically similar examples over web-scale text and image datasets, demonstrating significant speed up in convergence for downstream tasks.

For genomic semantic deduplication, we first trained a 150M gLM2 on the tokenized OMG dataset for 600k steps. We then embed the entire OMG dataset, by extracting a mean-pooled, per-example representation from the model's last hidden layer. The example-level embeddings correspond closely to the taxonomic classification available for the OG dataset (Fig. 1A). This motivates embedding-based deduplication as a method for removing near duplicates while balancing taxonomic bias. We prune the OMG dataset at 49% (i.e. 49% of the original data is removed) at the deduplication threshold 2e-3 (where examples with embeddings <2e-3 in cosine distance are deduplicated) (Appendix G). The pruned examples are saturated in highly dense clusters (Fig. 1B) which results in taxonomic balancing (Appendix H) , measured by increased entropies of distribution across taxonomic levels (Appendix I). We then trained a 150M gLM2 on the pruned OMG dataset for an equal number of steps, and compared its performance against the un-pruned version on DGEB. While pruning results in a modest increase in the aggregate DGEB score (0.48 vs 0.47), we observe improvements in tasks that feature underrepresented taxa (e.g. ArchRetrieval, RpoB Arch phylogeny) (Appendix J). This improved performance for underrepresented taxa appears to come at the cost of small regressions on tasks that are biased towards overrepresented taxa. Genomic SemDeDup presents a tunable method for effectively pruning unstructured genomic data without reliance on taxonomic labels.

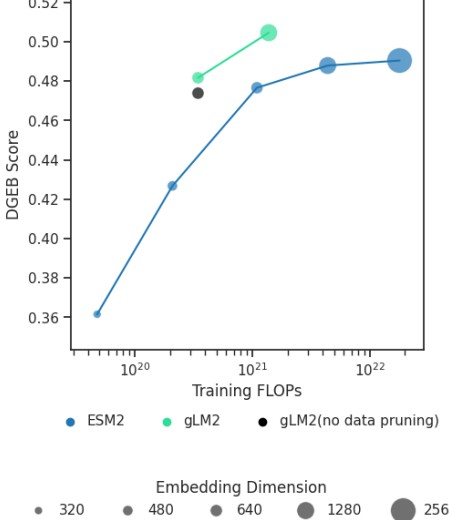

Figure 3: **Scaling performance on DGEB amino acid tasks for gLM2 and ESM2, relative to pretraining floating point operations (FLOPs).** gLM2_150M trained with no data pruning is shown in black.

## 4.3 GLM2 PERFORMANCE ON DGEB

We compare the performance of the 150M and 650M gLM2 models trained on the pruned OMG dataset against the ESM2 series trained on the UniRef50/D dataset (Fig. 3). gLM2 outperforms ESM2 on the overall DGEB score at each parameter scale. In particular, gLM2's performance scales with pretraining floating point operations (FLOPs) on protein tasks where ESM2 plateaus in performance with scaling (i.e. Operon pair classification tasks, ModBC paralogy task) (Appendix K). Such improved functional representation learning is likely due to gLM2's ability to leverage genomic context information, and thereby learn relationships between genomic elements. gLM2, being a mixed-modality model, also learns intergenic sequence representations. We compare gLM2's performance on DGEB nucleic acid (NA) tasks against the Nucleotide Transformer series (Appendix L). gLM2 performs similarly on NA tasks when compared to Nucleotide Transformers, despite only a small fraction of the training tokens consisting of DNA sequences.

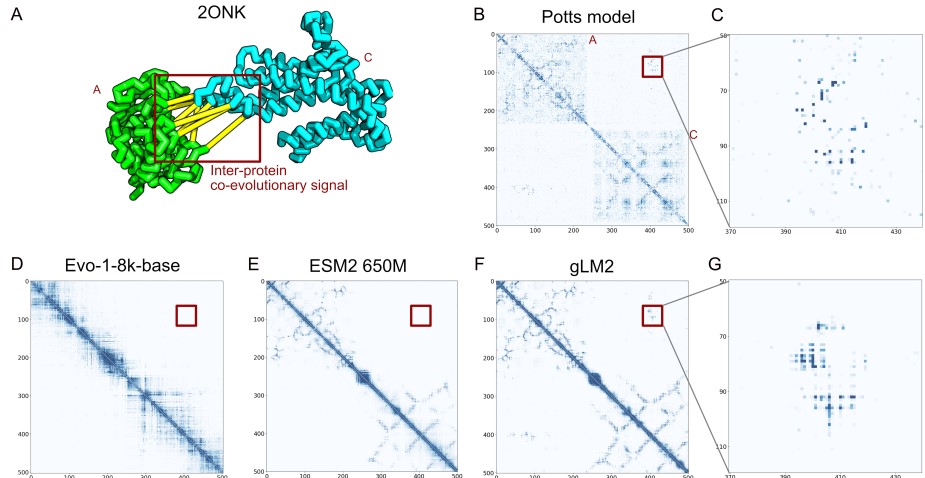

Figure 4: **gLM2 learns protein-protein interface co-evolutionary signal in the 2ONK (ModBC) complex.** **(A)** ModB (Chain C) and ModC (Chain A) forms a structural complex with co-evolutionary signal between residues (in yellow). **(B)** Co-evolutionary signal extracted from multiple sequence alignment of 2ONK[4](Ovchinnikov et al., 2014), calculated and visualized using GREMLIN (`PDB_benchmark_alignments/2ONK_A2ONK_C.fas`). The region of inter-protein co-evolutionary signals are highlighted with a red box. **(C)** Zoomed-in region of inter-protein coevolutionary signal in B. **(D)** Categorical Jacobian calculated for Evo on the DNA sequence encoding 2ONK_A and 2ONK_C (from 89,891 to 91,376 of genomic sequence NC_000917.1). The L2 norm was computed over the (3,4,3,4) tensor for every pair of codon positions to generate the contact map. **(E)** Categorical Jacobian calculated for ESM2 650M on the concatenated 2ONK_A_2ONK_C sequence. No inter-protein co-evolutionary signal is detected. **(F)** Categorical Jacobian calculated for gLM2_650M on the concatenated 2ONK_A_2ONK_C sequence. **(G)** Zoomed-in region of inter-protein coevolutionary signal in F.

## 4.4 GLM2 LEARNS PROTEIN-PROTEIN INTERACTION INTERFACES

We test gLM2's ability to learn coevolutionary signals between proteins in protein-protein interaction interfaces (Ovchinnikov et al., 2014). Previous studies have shown that pLMs learn within-protein co-evolutionary information that can be extracted with a supervised contact prediction head (Lin et al., 2023) using an unsupervised "categorical Jacobian" calculation (Zhang et al., 2024). However, pLMs trained on individual proteins or protein families cannot learn co-evolutionary information across proteins. We calculate the categorical jacobian values from gLM2_650M on the concatenated sequence of 2ONK_A (ModC)

and 2ONK_C (ModB) (Appendix N). We demonstrate that gLM2 leverages multi-protein context to learn protein-protein interfaces from a single concatenated sequence that closely matches the co-evolutionary signal that can be learned from multiple sequence alignment (MSA) based Potts model (GREMLIN (Kamisetty et al., 2013)) (Fig. 4). Such protein-protein interface signals cannot be extracted in existing language model methods such as ESM2 650M and Evo-1-8k-base (Fig. 4E and F). We validate the gLM2-predicted contacts directly with the ground truth contacts from 2ONK PDB structure (Fig. 5), as well as 31 complexes previously described in (Ovchinnikov et al., 2014) (Appendix **??**). The ability to extract interacting residues without supervision nor MSA presents an opportunity to predict novel protein-protein interactions from sequence information alone.

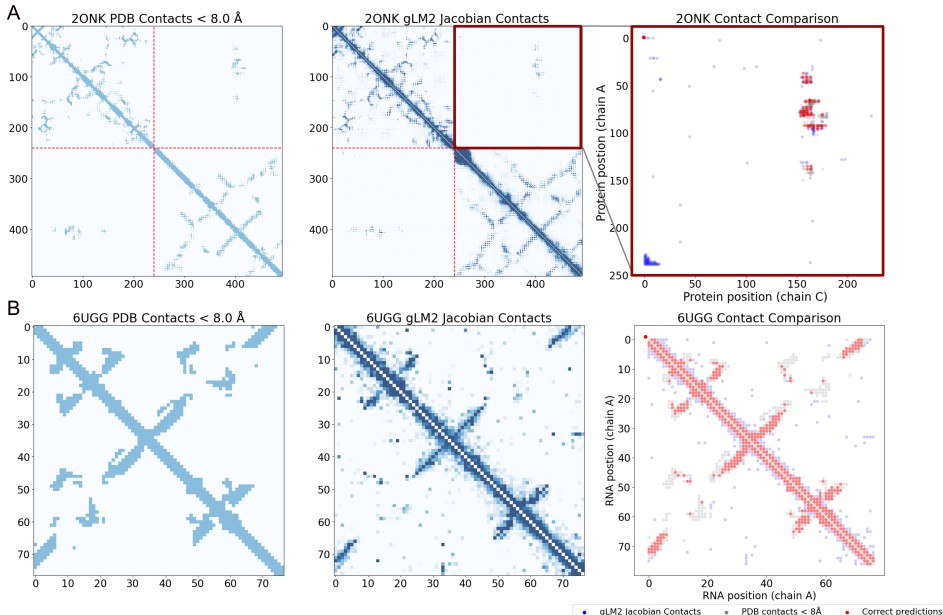

Figure 5: **Ground truth comparisons of Jacobian-detected contacts against PDB structures**. **(A)** Left: Ground truth contacts derived from PDB structure (PDB: 2ONK; ModBC complex) shown in Fig. 4, where contact is defined as residues that are within <8Å. Middle: gLM2-predicted contacts using Categorical Jacobian. Right: Inter-protein region highlighting top n highest scoring predicted contacts (red for true positive, blue for false positive) overlaying ground truth contacts (gray), where n is the number of inter-protein contacts identified in the ground truth. **(B)** Left: Ground truth contacts derived from tRNA-Asp (PDB: 6UGG) shown in Fig. 6. Middle: gLM2-predicted contacts using Categorical Jacobian. Right: Top n highest scoring contacts in gLM2 (red for true positive, blue for false positive) overlaying ground truth contacts (gray), where n is the number of contacts within tRNA identified in the PDB ground truth excluding the diagonal.

### 4.5   GLM2 LEARNS REGULATORY SYNTAX IN INTERGENIC DNA

We demonstrate gLM2's ability to identify regulatory syntax and non protein-coding elements in IGS regions. We first validate gLM2's ability to predict contacts in tRNA-Asp against the ground truth 6UGG PDB structure (Fig. 5) We further demonstrate gLM2's ability to identify regulatory regions (sigma factor binding and terminator) in the genomic context of tRNA-Asp (Fig. 6). We additionally observe a signal downstream of *aspV* and upteam of the terminator region. This region lacks annotation in EcoCyc (Karp et al., 2023) and presents the potential for gLM2-based unsupervised discovery of novel regulatory sequence motifs. We examined 23 additional intergenic regions in the *E. coli* K-12 genome that

---

[4]`https://colab.research.google.com/github/sokrypton/GREMLIN_CPP/blob/master/GREMLIN_TF.ipynb`

contain at least one terminator and one promoter regions according to EcoCyc annotations. We show conserved Categorical Jacobian patterns corresponding to previously validated annotations across diverse regions of the genome (Appendix P). We further conducted a similar analysis on *B. subtilis* 168 genomic region 119,848-120,978bp (5'->3') containing a L10 leader RNA gene and two ribosomal protein coding genes *rplJ* and *rplL* (Appendix O). We observe putative contacts between the L10 leader RNA and ribosomal protein RplL, an experimentally evidenced interaction (Johnsen et al., 1982). We also observe contacts between RplJ and RplL, a known ribosomal protein complex. Furthermore, our analysis highlights co-evolutionary signal between the Shine-Dalgarno sequences (ribosomal binding site) upstream of *rplJ* and *rplL*, suggesting gLM2 understanding of genome-specific regulatory motifs.

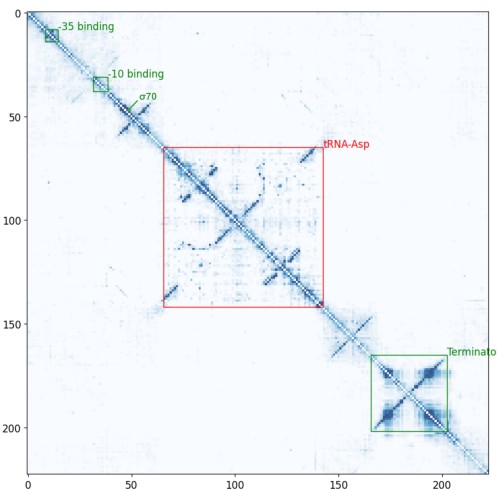

Figure 6: **gLM2 learns intergenic regulatory syntax and tRNA structure.** We visualize co-evolutionary signal in *E. coli* K-12 substr. MG1655 chromosomal region 236,866-237,087bp (5'->3') containing *aspV* (tRNA-Asp encoding gene) using the Categorical Jacobian. Structural signatures in tRNA-Asp sequence are visible. Other signals correspond to known regulatory syntax including sigma factor binding sites (-35 and -10), transcription initiation site ($\sigma_{70}$ binding region), and rho-independent terminator sequence.

## 5 CONCLUSION

The OMG dataset is a large-scale mixed-modality biological pretraining corpus that leverages the immense volume and diversity of unstructured metagenomic (primarily prokaryotic and viral) sequences. We quality-filter and preprocess the raw metagenomic sequences into a mixed-modality format ready for language model training. We showcase the efficacy of mixed-modality input for genomic language modeling with gLM2. With genomic SemDeDup, we present an efficient method for reducing the bias and duplication in genomic datasets. The gLM2 models trained on pruned OMG learn contextualized representations for both CDS and IGS, and demonstrate efficient scaling and improved performance across downstream tasks compared to uncontextualized protein language models trained on curated databases. We further demonstrate the gLM2's ability to learn protein-protein interfaces at residue-level, paving the path towards unsupervised protein-protein complex prediction. Finally, we show that gLM2 learns evolutionary couplings of regulatory motifs in the intergenic DNA, indicating model understanding of both modalities of the data. The OMG dataset and gLM2 models as well as the supporting code are publicly available for download.

ETHICS STATEMENT

This study aims to advance open science for genomics, by making the OMG corpus and gLM2 model publicly available on the HuggingFace Hub. The OMG corpus is constructed from publicly available data within JGI's IMG and EMBL's MGnify repositories. We exclude all embargoed and restricted data from the OMG corpus. As the data originates from environmental samples, no personally identifiable information is associated with the dataset.

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

## APPENDIX A    DATA SOURCES

|       | Type        | Snapshot date | # Samples | # contigs* | Total bps | # CDS |
|-------|-------------|---------------|-----------|------------|-----------|-------|
| IMG   | Metagenomes | 2023-08-27    | 36,273    | 182M       | 1.70T     | 1.84B |
|       | Genomes     | 2023-08-27    | 131,744   | 6.2M       | 0.4T      | 0.4B  |
| MGnify | Metagenomes | 2022-11-23   | 33,531    | 82M        | 1.03T     | 1.03B |

*Number of contigs after filtering and preprocessing.

## APPENDIX B    DATASET PREPROCESSING

Sequences (purple) undergo filtering steps (green), yielding three Hugging Face datasets (yellow) made available with this paper. 'NA' and 'AA' refer to nucleic acid and amino acid data modalities respectively.

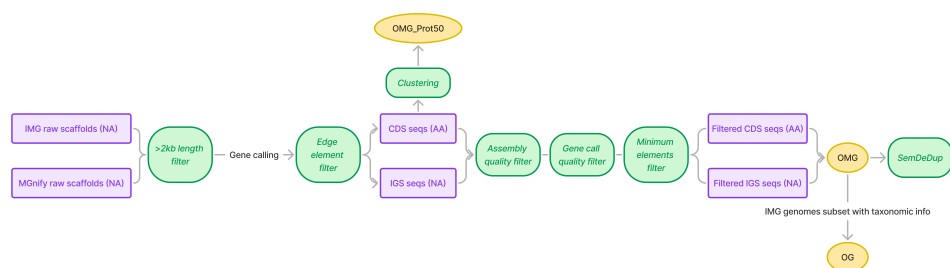

## APPENDIX C    DATASET LENGTH DISTRIBUTIONS

Length distributions of the OMG corpus. **(A)** Distribution of contig lengths in the number of genomic elements (CDS and IGS). **(B)** Distribution of contig lengths in base pairs. **(C)** Distribution of CDS lengths in amino acids. **(D)** Distribution of IGS lengths in base pairs.

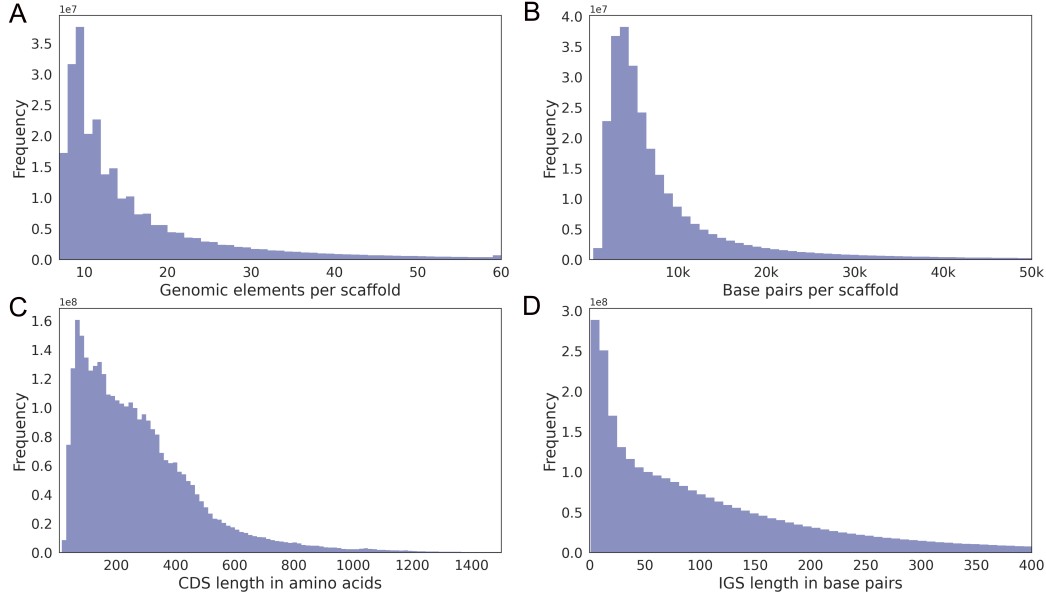

## Appendix D    Invalid Character Distributions

Distribution of the percent of characters per genomic element considered as invalid ("X" for amino acids and "N" for nucleotides) prior to applying the assembly quality filter from Section 3.1. The assembly quality filter removes elements containing more than 20% invalid characters, resulting in 0.004% of CDS and 0.2% of IGS being filtered from OMG. We show the distribution for the subset of genomic elements containing at least 1 invalid character.

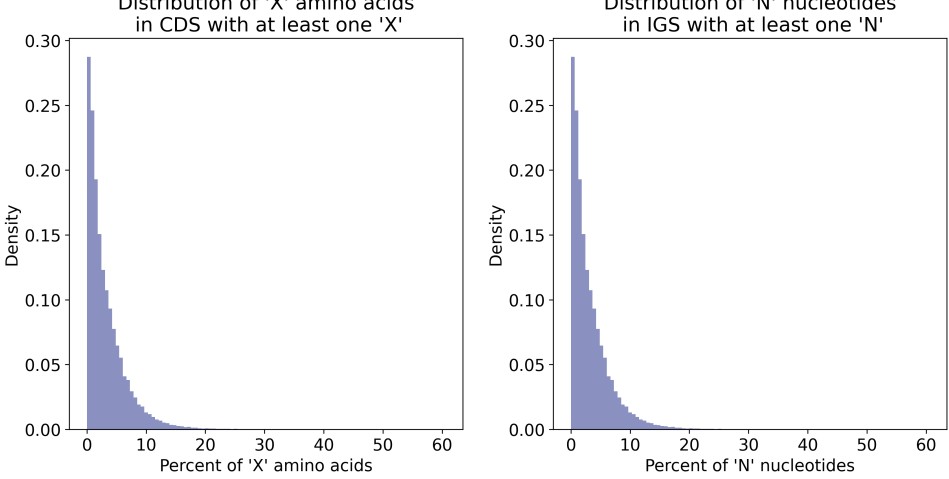

## Appendix E    OMG_prot50 clustering method

A total of 4.2B protein sequences were first clustered to remove fragments using MMseqs2 linclust (Steinegger & Söding, 2018) (commit f6c98, parameters:–min-seq-id 0.9 -c 0.9 –cov-mode 1). Subsequently, the resulting sequences were clustered at 50% sequence id and 90% sequence coverage using MMseqs2 `linclust -min-seq-id 0.5 -c 0.9`. Singleton clusters (only one sequence in the cluster across the full dataset) were removed and remaining 207M cluster representatives were uploaded as the Hugging Face dataset.

## APPENDIX F   GLM2 MODEL PARAMETERS

gLM2 is a transformer encoder optimized using AdamW (Loshchilov & Hutter, 2019) and trained in mixed precision bfloat16. We set the AdamW betas to (0.9, 0.95) and weight decay of 0.1. We disable dropout throughout training. The learning rate is warmed up for 1k steps, followed by a cosine decay to 10% of the maximum learning rate. gLM2 uses RoPE (Su et al., 2023) position encoding, SwiGLU (Shazeer, 2020) feed-forward layers, and RMS normalization (Zhang & Sennrich, 2019). We leverage Flash Attention 2 (Dao, 2023) to speed up attention computation over the sequence length of 4096.

|  | Dim | Num heads | Num layers | Context length | Learning rate | Batch size | Pretraining tokens |
|---|---|---|---|---|---|---|---|
| gLM2-150M | 640 | 10 | 30 | 4096 | 1e-3 | 128 | 315B |
| gLM2-650M | 1280 | 20 | 33 | 4096 | 1e-3 | 128 | 315B |

## APPENDIX G    SEMANTIC DEDUPLICATION DISTANCE THRESHOLD

The percentage of remaining training examples as a function of the embedding distance threshold. Examples within the distance threshold in embedding space are deduplicated.

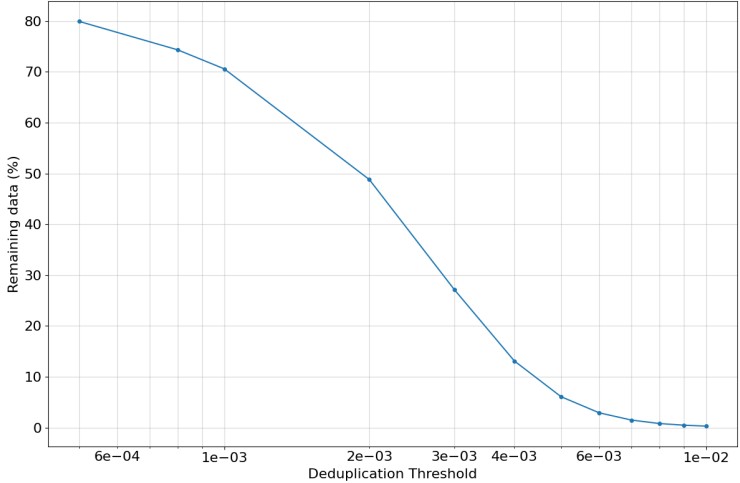

## APPENDIX H    TAXONOMIC DISTRIBUTION OF THE OG DATASET BEFORE AND AFTER PRUNING

Data pruning through semantic deduplication reduces dataset bias toward overrepresented phyla and orders.

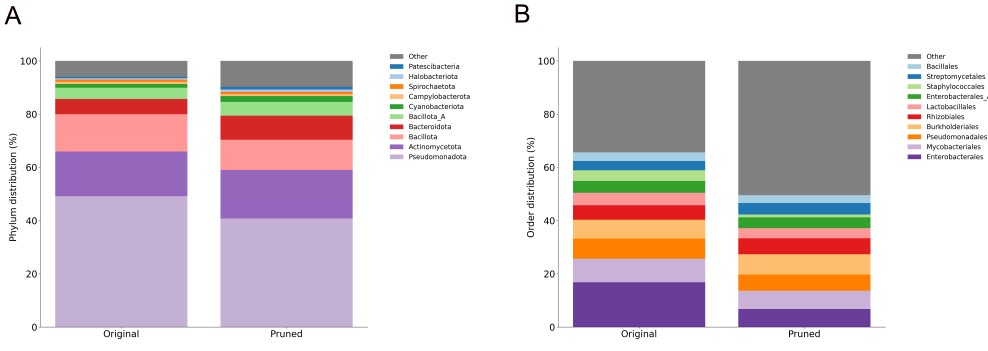

## APPENDIX I   TAXONOMIC ENTROPY OF THE OG DATASET BEFORE AND AFTER PRUNING

Semantic deduplication of the OG dataset consistently increases the taxonomic entropy across all taxonomic ranks, indicating a more even distribution.

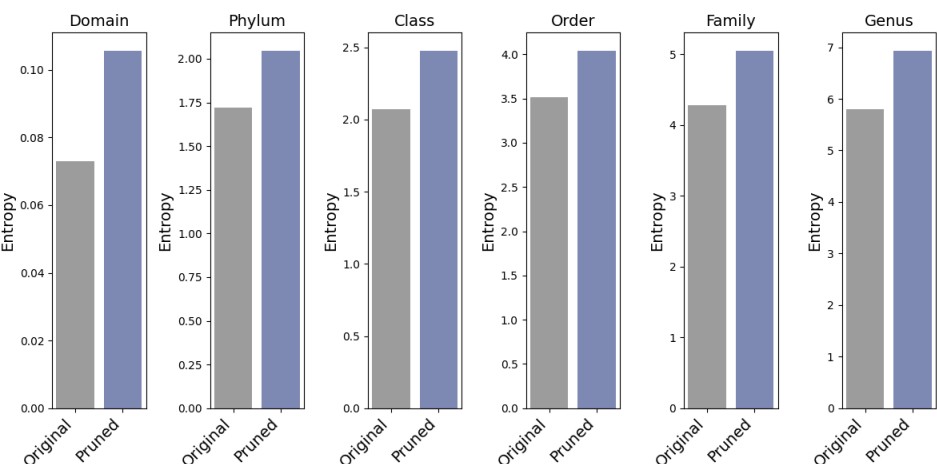

# APPENDIX J  ABLATION OF SEMANTIC DEDUPLICATION

We train two 150M parameter gLM2 models on the original and pruned OMG dataset, each for 600k steps. Both models are evaluated on the DGEB benchmark. Pruning improves performance, especially for tasks with under-represented sequences.

| | Bac Arch BiGene | Arch Retrieval | Cyano Oper. Retrieval | ModBC Paralogy BiGene | FeFe Hydrogenase Phylogeny | RpoB Archaeal Phylogeny | EC Classification | MIBiG Classification | E.coli Operonic Pair | Euk Retrieval | Convergent Enzymes Classification | MopB Clustering | Vibrio Operonic Pair | RpoB Bacterial Phylogeny | DGEB Score |
|---|---|---|---|---|---|---|---|---|---|---|---|---|---|---|---|
| **gLM2_150M** (no data pruning) | **0.750** | 0.289 | 0.408 | **0.244** | 0.688 | 0.312 | **0.517** | **0.670** | 0.628 | **0.348** | **0.196** | 0.786 | 0.547 | 0.251 | 0.474 |
| **gLM2_150M** (with data pruning) | 0.722 | **0.294** | **0.418** | 0.229 | **0.716** | **0.353** | 0.514 | 0.660 | **0.639** | 0.330 | 0.172 | **0.820** | **0.566** | **0.311** | **0.482** |

## Appendix K   Per task DGEB scaling with FLOPs for ESM2 and gLM2 models in amino acid tasks

Primary metric from the best scoring layer (between mid, and last) is reported for each task. To account for model-specific patterns in learning task-relevant functional information across different layers in the network (West-Roberts et al., 2024), DGEB calculates model performance for both mid and last layer and reports the best score between the two.

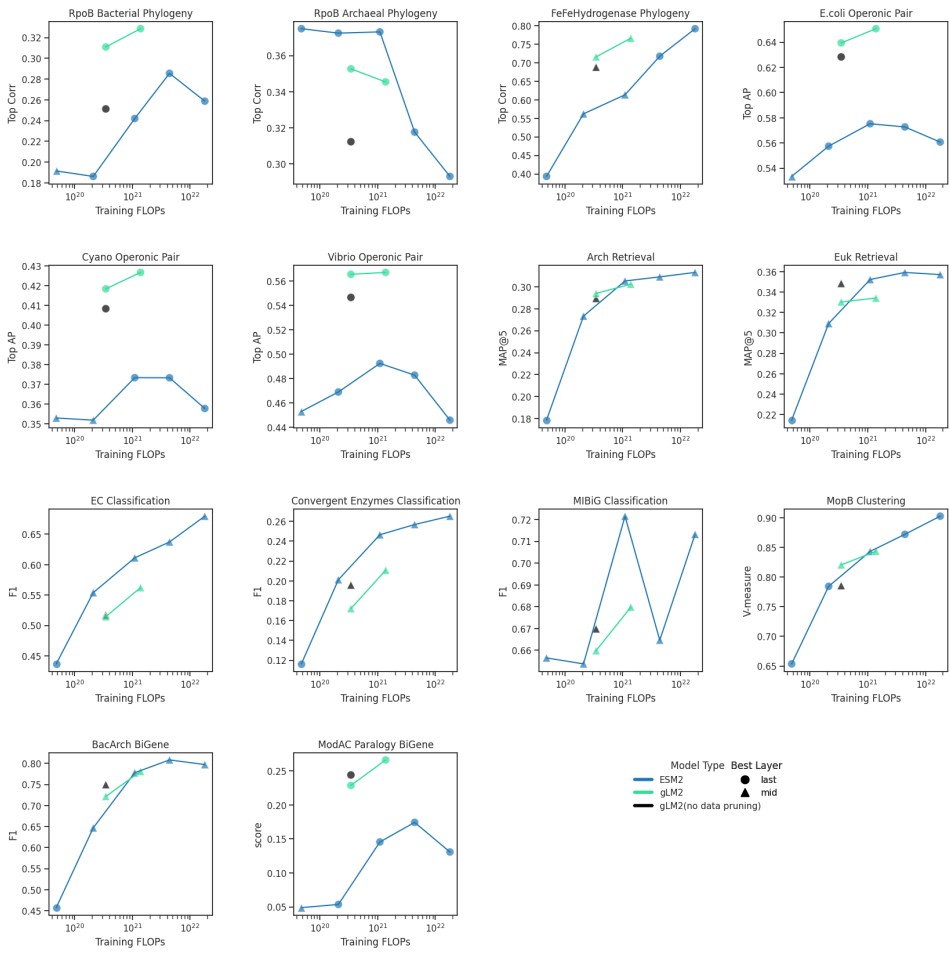

## Appendix L    Per task DGEB scaling with FLOPs for Nucleotide Transformers and gLM2 models in nucleic acid tasks.

Primary metric from the best scoring layer (between mid, and last) is reported for each task. To account for model-specific patterns in learning task-relevant functional information across different layers in the network (West-Roberts et al., 2024), DGEB calculates model performance for both mid and last layer and reports the best score between the two.

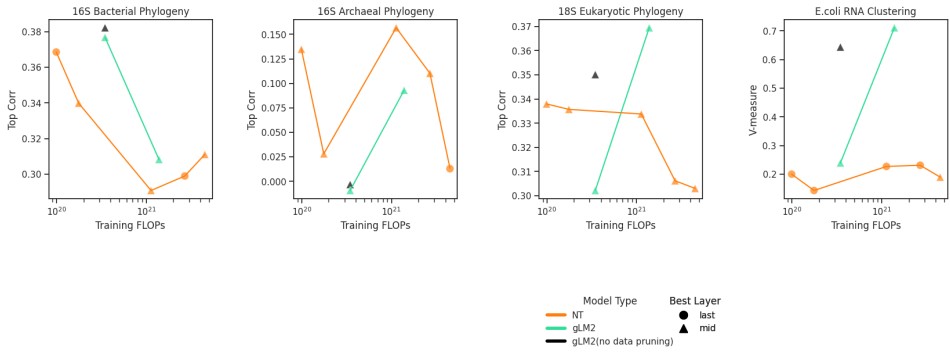

## Appendix M    GLM2 performance on ProteinGym

| Model name | Avg Spearman | Activity | Binding | Expression | Organismal Fitness | Stability |
|---|---|---|---|---|---|---|
| ESM2_650M | 0.414 | 0.425 | 0.337 | 0.415 | 0.369 | 0.523 |
| gLM2_650M_prot | 0.384 | 0.406 | 0.327 | 0.412 | 0.311 | 0.466 |

We evaluate gLM2 on the ProteinGym (Notin et al., 2023) Deep Mutational Scanning (DMS) substitutions task. Because the DMS task is strictly a single-protein task (without context), we benchmark gLM2_650M after finetuning for one epoch of OMG_prot50, the single-protein dataset introduced in Table 1. While gLM2_650M_prot performs slightly worse than ESM2_650M, we note that the ProteinGym benchmark includes eukaryotic sequences, which are poorly represented in the OMG dataset.

## Appendix N    ModB and ModC sequence concatenation

This concatenated sequence was derived from the 2ONK_A_2ONK_C alignment used in (Ovchinnikov et al., 2014).

```
MFLKVRAEKRLGNFRLNVDFEMGRDYCVLLGPTGAGKSVFLELIAGIVKPDRGEVRLNGADITPLPPPERGIGFV
PQDYALFPHLSVYRNIAYGLRNVERVERDRRVREMAEKLGIAHLLDRKPARLSGGERQRVALARALVIQPRLLLLDEPLSAV
DLKTKGVLMEELRFVQREFDVPILHVTHDLIEAAMLADEVAVMLNGRIVEKGKLKELFSAKNGEVAEFLSARNLLLKVSKIL
DMRLLFSALLALLSSIILLFVLLPVAATVTLQLFNFDEFLKAASDPAVWKVVLTTYYAALISTLIAVIFGTPLAYILARKSF
PGKSVVEGIVDLPVVIPHTVAGIALLVVFGSSGLIGSFSPLKFVDALPGIVVAMLFVSVPIYINQAKEGFASVDVRLEHVAR
TLGSSPLRVFFTVSLPLSVRHIVAGAIMSWARGISEFGAVVVIAYYPMIAPTLIYERYLSEGLSAAMPVAAILILLSLAVFV
ALRIIVGREDVSEGQG
```

## APPENDIX O    PUTATIVE RNA-PROTEIN-PROTEIN INTERACTIONS

We visualize a contiguous stretch (119,848-120,978bp, 5'->3') of the *B. sutilis* 168 reference genome. Putative residue-level interactions between L10 leader RNA (*ldlJ*), proteins RplJ and RplL are highlighted in gray boxes. Shine-Dalgarno sequences upstream of the two protein-coding genes are highlighted and co-evolve.

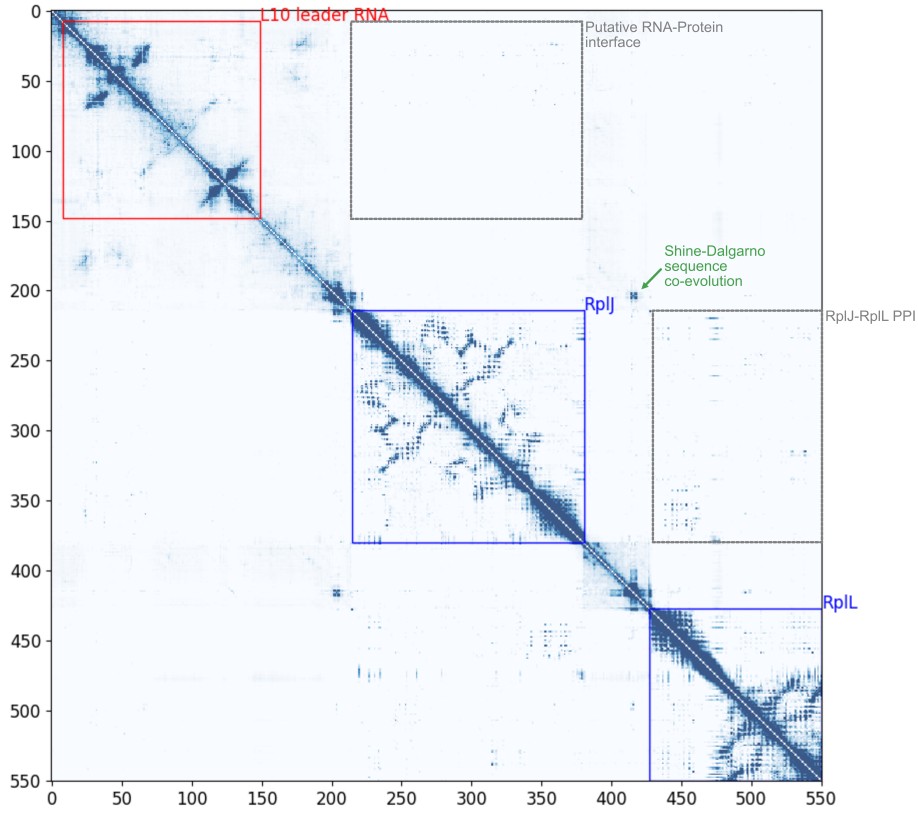

## APPENDIX P    ADDITIONAL FILES

Additional Files are found in https://zenodo.org/records/14198868

**Additional File 1.** OG sample ID to original NCBI metadata. A JSON file mapping OG sample ID (taxon_oid) to NCBI metadata (accessions, collection dates).

**Additional File 2.** DOIs for MGnify samples. DOIs for MGnify samples that were included in this study, where available.

**Additional File 3.** DOIs for IMG samples, DOIs for IMG samples that were included in this study, where available. **Additional File 4.** Comparison of gLM2 Jacobian Contacts on 2ONK with (A) and without (B) the 2 basepair IGS sequence flanking ModB and ModC. We show that the addition of IGS sequence does not change the results. **Additional File 5.** A zip file containing all 32 evolutionary conserved complexes in PDB previously identified in (Ovchinnikov et al., 2014), `https://openseq.org/cplx.php?sort=prob&order=DESC&mode=pdb`. PDB contacts and gLM2 Jacobian Contacts are compared.

**Additional File 6.** A zip file containing Categorical Jacobian maps of 26 IGS regions in *E.coli* K-12 str. MG1655 (Genome ID: U00096) with at least one promoter (highlighted in red) and one terminator (highlighted in green) sites annotated in EcoCyc. File names and figure title correspond to the start and end positions in the U00096 genome.

