# OpenReview forum: "The OMG dataset: An Open MetaGenomic corpus for mixed-modality genomic language modeling"
_ICLR.cc/2025/Conference — ICLR 2025 Poster_

### Official Review · Reviewer_GgLu · 2024-11-01

**Soundness:** 3
**Presentation:** 3
**Contribution:** 3
**Rating:** 5
**Confidence:** 3

**Summary:**

The authors present the Open MetaGenomic (OMG) dataset, a substantial, mixed-modality genomic corpus designed to support biological language model training. OMG is sourced from two major repositories, JGI’s IMG and EMBL’s MGnify, encompassing 3.1 trillion base pairs and 3.3 billion protein-coding sequences. The authors address critical challenges in metagenomic data usage, including accessibility, quality filtering, and deduplication. They also introduce gLM2, the first mixed-modality genomic language model pretrained on OMG with 150M and 650M parameters, capable of leveraging both genomic and protein-coding information to capture complex functional and regulatory patterns. gLM2 demonstrates decent improvements in functional representation learning of both protein and NA and co-evolutionary information modeling, outperforming existing models on tasks involving genomic structure prediction, protein-protein interactions, and regulatory syntax learning.

**Strengths:**

1. **High-Quality, Diverse Metagenomic Dataset**: By aggregating two of the largest metagenomic repositories, the authors create a richly diverse dataset that addresses critical gaps in sequence variety and covers underrepresented taxa through rigorous filtering and preprocessing. This dataset is likely to become an essential training resource for future models, representing a major contribution to the field.

2. **Mixed-Modality Model**: The authors propose an innovative modeling approach that integrates both protein and nucleic acid sequences, enabling the model to process protein-coding and regulatory sequences together. This combined approach opens the door to unique and complex downstream applications.

3. **Co-Evolutionary Signals**: This work may be the first biomolecular language model to demonstrate coevolutionary signals across sequences using the Categorical Jacobian method, marking a novel and compelling contribution to biomolecular modeling.

**Weaknesses:**

Since I don't have much experience in metagenomic and its relevant data processing, I will leave these aspects to other reviewers. Here, I will focus more on gLM2 model and performance.

1. The model is evaluated on the DGEB benchmark. However, insufficient information is provided about the evaluation methodology and associated metrics. I recommend adding a section discussing the evaluation process and metrics in detail.

2. After checking the original DGEB paper, it appears that gLM2 can not match ProGen2 on amino acid (AA) tasks (Appendix J), which is not reported in the current manuscript. The authors should either include these results or justify their omission.

3. From a protein language model perspective, the evaluation needs to be more comprehensive. The authors should include evaluations on standard benchmarks such as FLIP or ProteinGym.

4. For the coevolution signal extraction experiment with 2ONK, the comparison would benefit from including NT2 and ProGen2 as additional baselines.

5. While sections 4.4 and 4.5 present well-analyzed individual cases, for contact prediction in particular, more examples are needed to demonstrate that gLM2 consistently outperforms alternative methods. Might be interesting to consider datasets like [1][2] for RNA and  CAMEO for protein.


[1] https://github.com/jaswindersingh2/SPOT-RNA-1D
[2] https://pmc.ncbi.nlm.nih.gov/articles/PMC7297115/

**Questions:**

1. Regarding the training and modeling capabilities of gLM2:
   - Is gLM2 trained on contig sequences containing both nucleotide sequences (IGS) and protein sequences? If so, does this limit the model to only modeling interactions between bio-sequences from the same contig? Can gLM2 predict contacts for Protein hetero-multimer interactions for example (also RNA-Protein interactions and DNA-Protein interactions)?


2. For the 2ONK example:
     * Are chains A and C from the same contig?
     * If yes, is there IGS between them?
     * What would be the effect on coevolution signal extraction if IGS were included?
     * Why were only chains A and C analyzed when 2ONK has 4 chains?

3. How ESM2 FLOPs were estimated? Or if this information comes from existing literature (e.g., the original ESM2 paper).

4. Regarding model hyperparameter design:
   - What is the rationale behind the chosen head number, layer number, and embedding dimensions?
   - Were preliminary experiments conducted to support these choices?
   - Contemporary LLMs like LLaMA typically use equal numbers of heads and layers, with embedding dimensions set to 128 times the number of heads

---

> ### Author Response · Authors · 2024-11-21
> **Response to Reviewer GgLu**
>
> Thank you for your thoughtful review. We are glad that you find OMG as "a major contribution to the field" and that gLM2 protein-protein coevolution marks a "novel and compelling contribution to biomolecular modeling".
>
> > Insufficient information is provided about the evaluation methodology and associated metrics.
>
> Thank you for this suggestion. We have added the following sentences to Section 4.1: "We benchmark gLM2 on the Diverse Genomic Embedding Benchmark (DGEB) (West-Roberts, 2024).  DGEB is a comprehensive benchmark that evaluates model representations across diverse taxa and 18 tasks representing multiple axes of biological function, such as evolutionary distance similarity, remote homology prediction, enzyme classification, and retrieval sensitivity."
>
> > gLM2 can not match ProGen2 on amino acid (AA) tasks (Appendix J), which is not reported in the current manuscript. The authors should either include these results or justify their omission.
>
> The primary focus of this paper is to motivate OMG as a pre-training dataset. We focus our comparison with the ESM2 series to minimize conflating other factors such as Progen2's autoregressive architecture and causal language modeling task. We design gLM2 to closely match ESM2 (same number of layers, hidden dimensions, pre-training objective). In order to include Progen2 as an appropriate baseline, we would need to train an autoregressive model on OMG, which is beyond the scope of this study.
>
> > The authors should include evaluations on standard benchmarks such as FLIP or ProteinGym.
>
> Thank you for the suggestion. We have added ProteinGym results in Appendix M. Because the DMS task is strictly a single-protein task (without context), we benchmark gLM2_650M after fine-tuning for one epoch of OMG_prot50 dataset (Table 1) and we find that gLM2 performs slightly worse than ESM2 on Avg Spearman score (0.384 vs 0.414).  We note that the ProteinGym benchmark includes eukaryotic sequences, which are poorly represented in the OMG dataset, and likely impacts gLM2 performance.
>
> > For the experiment with 2ONK, the comparison would benefit from including NT2 and ProGen2.
>
> We choose ESM2 and Evo as pLM and gLM baselines because the Categorical Jacobian relies on per amino-acid/nucleotide resolution, which NT models do not have due to k-mer tokenization. We expect Progen2 to perform similarly to ESM2 as neither has been trained on multi-gene contexts.
>
> > While sections 4.4 and 4.5 present well-analyzed individual cases, for contact prediction in particular, more examples are needed to demonstrate that gLM2 consistently outperforms alternative methods.
>
> To address the reviewer's feedback of presenting individual cases on Section 4.4, and 4.5, we added supplementary information expanding on these two analyses in Appendix P, Additional File 5 and Additional File 6.
>
> > Is gLM2 trained on contig sequences containing both nucleotide sequences (IGS) and protein sequences?
>
> Yes, gLM2 is trained on multi-modal contig sequences with up to 4096 tokens. This limits modeling between proteins in the same context (in the case of microbial systems, interactions often occur with co-located genes).
>
> > Can gLM2 predict contacts for Protein hetero-multimer interactions (also RNA-Protein interactions and DNA-Protein interactions)?
>
> The 2ONK example from Figure 4 is an example of hetero-multimer prediction using gLM2. For homo-multimers, we were not able to decouple inter-chain interactions from intra-chain interactions in Categorical Jacobian analysis of a single sequence. This could be an interesting avenue for future work. For RNA/DNA-protein interactions, we showcase a putative RNA-protein interaction in Appendix O, however there currently exists no structural groundtruth for the L10 leader RNA and RplJ complex. In our future work, we hope to follow up on RNA/DNA-protein interactions.
>
>
> > How ESM2 FLOPs were estimated?
>
> ESM2 pretraining flops were computed in xTrimoPGLM (https://arxiv.org/pdf/2401.06199) Table 4 (page 83).
>
>
> > Questions regarding model hyperparameter design
>
> As mentioned above, we choose architecture parameters to match the ESM2 series.

---

> ### Comment · Reviewer_GgLu · 2024-11-23
>
> Thank you to the authors for your responses. I appreciate the effort in curating such the dataset for the community, which I anticipate will be useful. However, while the paper claims its primary objective is to introduce the OMG dataset, over half of the content discusses the gLM2 model. This seems contradictory to me. Given the current structure of the paper, I believe the gLM2 model requires more rigorous evaluation and should significantly influence the review scores. If the focus is intended to be on the dataset, I suggest reformatting the paper.

---

> > ### Author Response · Authors · 2024-11-23
> > **Response to Reviewer GgLu**
> >
> > Thank you for your feedback and for acknowledging the value of the OMG dataset.
> >
> > > However, while the paper claims its primary objective is to introduce the OMG dataset, over half of the content discusses the gLM2 model. This seems contradictory to me.
> >
> > We understand your concern regarding the perceived emphasis on the gLM2 model in our manuscript. We would like to clarify that our primary objective is indeed to introduce the OMG dataset, and the gLM2 model serves as a crucial tool to demonstrate its utility and potential impact.
> >
> > This follows established practices in the field when introducing large-scale datasets, and of particular importance within biological domains and novel data formats, where the introduction of a model is necessary to provide tangible evidence of the dataset's value.
> >
> > Notable examples of large scale datasets motivated by a new model include:
> >
> > 1. The Pile [1], which introduces "a 825.18 GiB english language dataset for language modeling", and demonstrates "significant improvements across many domains by GPT-2- sized models trained on this new dataset".
> >
> > 2. The FineWeb Datasets [2] , which introduces "FineWeb-Edu, a 1.3-trillion token collection of educational text" and demonstrates that "LLMs pretrained on FineWeb-Edu exhibit dramatically better performance on knowledge- and reasoning-intensive benchmarks".
> >
> > Similarly, in our manuscript, the gLM2 model serves to validate OMG's value as a pretraining dataset. Thanks to reviewer suggestions, we have addressed concerns regarding gLM2's evaluation by expanding the evaluation to multiple benchmarks (DGEB and ProteinGym) and expanding analysis for the novel tasks (PPI and regulatory syntax).
> >
> > We also want to emphasize that significant portions of our paper are dedicated to describing the dataset itself. Specifically, Section 3 details each step of the dataset curation and filtering process, Section 4.2 details the deduplication process, and Appendix A through E provide details on dataset composition and preprocessing steps.
> >
> > However, we appreciate your feedback and are willing to further clarify the manuscript to ensure the primary focus on the dataset is more explicit.
> >
> >
> >
> > [1] https://arxiv.org/abs/2101.00027
> >
> > [2] https://arxiv.org/abs/2406.17557

---

### Official Review · Reviewer_Sobx · 2024-11-01

**Soundness:** 4
**Presentation:** 4
**Contribution:** 4
**Rating:** 8
**Confidence:** 4

**Summary:**

The Open Megagenomic Corpus (OMG) and gLM2 model are important and novel contributions to biologic sequence modeling. Authors make use of the largest repositories of metagenomic data to construct a new database for training that is larger than exiting, popular biologic LM training datasets. Using both coding sequences and intergenic sequences, authors show that their modeling strategy captures long range interactions of coding and non-coding genomic elements at multiple scales of biologic sequence interaction.

The paper should be accepted given the technical and strategic innovation database creation and the novelty in modeling strategy that performs on par with an existing state-of-the-art model and captures important biology across sequence modalities.

**Strengths:**

The work addresses the need for larger databases of sequence data that can be used to train language models for biologic applications by making use of rapidly growing repositories of metagenomic sequence data. Authors utilize knowledge and methods from recent insights in NLP to construct the large database used for training and develop their own genomic deduplication procedure that captures biologic reality. They also utilize deep domain knowledge about metagenomic sequencing to filter terabytes of raw data for database construction. The new curated database is made available to the community. The authors utilize a novel modeling approach, constituting of which includes multi-modal input in the form of amino acid and nucleotide data.

**Weaknesses:**

The language model trained does not explicitly include eukaryotic genomic sequences and it is unclear gLM2 is meant to be used to model eukaryotic sequences. Authors should be more explicit about the application for eukaryotic sequence modeling. Authors do mention contamination and likely inclusion of eukaryotic sequences in metagenomic samples but this in in the context of challenges and mitigation thereof with metagenomic sequence data.

Applications of the gLM2 model are not explored. It would be interesting to see performance of gLM2 in areas where more and diverse training data is of importance, such as remote homology, or where the multi-modal training strategy is helpful, such as contig-level prediction of taxonomy.

**Questions:**

Authors claim their modeling approach achieves "more efficient scaling" performance with flops on a set of benchmark tasks than a prior top performing model; however, they only show two training flops. When looking at Appendix J there are some tasks where scaling is observed and some where it does not. It would be a stronger claim if there were additional data points for gLM2. On a per-task basis, it appears that gLM2 has similar scaling to ESM2 even if at the aggregated score level the scaling looks slightly better (Figure 3).

Minor comments:
1. "start a new contig" should be "start a new element" (l.255, 266-267)
2. DGEB is not defined (l.333)
3. Figure 5 has two "n" that are mentioned but not present in the figure

---

> ### Author Response · Authors · 2024-11-21
> **Response to Reviewer Sobx**
>
> Thank you for your thorough review and feedback. We appreciate that you find OMG and gLM2 provides "important and novel contributions to biologic sequence modeling".
>
> > Authors should be more explicit about the application for eukaryotic sequence modeling.
>
> Thank you for the suggestion, we have added the following sentence as a limitation to metagenomic datasets (Section 1): "While pretraining on metagenomic datasets allows models to leverage rich molecular diversity and genomic context, these models are most suitable for microbial genomes and may result in out-of-distribution effects on eukaryotic sequences."
>
> > It would be interesting to see performance of gLM2 in areas where more and diverse training data is of importance, such as remote homology, or where the multi-modal training strategy is helpful, such as contig-level prediction of taxonomy.
>
>
> We agree that other applications of gLM2 could be explored, in addition to the applications we introduce for protein-protein interaction (Section 4.4) and regulatory syntax (Section 4.5). We note that the DGEB benchmark includes a diverse set of tasks; notably operon prediction, paralog matching, and phylogenetic distance prediction tasks, where training with genomic context boosts performance (Appendix K). We agree that taxonomic classification is an interesting application, in Fig 1A we show that gLM2 indeed learns distinct clustering of taxonomy via mean-pooled example-level representations without supervision, which motivates future exploration of this application.
>
> > Authors claim "more efficient scaling", however, they only show two training flops.
>
> We agree that the scaling efficiency claim is not sufficiently strong with 2 model scales, and we have removed this claim from Section 4.3.
>
> >  Minor comments...
> 1. We intended to say "start a new contig" as element refers to a protein (CDS) or intergenic sequence (IGS). When an invalid sequence is found, we remove it and start a new contig.
> 2. Thank you for pointing this out. We added a definition of DGEB in Section 4.1
> 3. The caption defines each "n" as the number of ground-truth contacts for each example.

---

> > ### Comment · Reviewer_Sobx · 2024-11-25
> >
> > Thank you for addressing these concerns. I would like to see the taxonomy question addressed more rigorously in the future.

---

### Official Review · Reviewer_HdWV · 2024-11-03

**Soundness:** 4
**Presentation:** 3
**Contribution:** 2
**Rating:** 6
**Confidence:** 4

**Summary:**

This paper reports a large, well-curated, database of genomic sequences (protein coding and nucleic acids, genomic and metagenomic) with utility as a genomic pretraining dataset for biologic modeling. The dataset is unique and the authors demonstrate its utility in training a novel mixed-modality genomic language model (gLM2) that leverages genomic context information to learn functional representations and coevolutionary signals in protein-protein interfaces and genomic regulatory syntax.  These are very different kinds of tasks and that the model performs well in the vignettes shown is very promising for both the OMG corpus and the gLM. The gLM is not well described or validated; the corpus itself (OMG) is the focus of the paper. I agree that OMG is a very useful resource for the community, particularly in combination with gLMs but the authors do not sufficiently validate their gLM aside from a few interesting but poorly described vignettes.

**Strengths:**

OMG is a novel resource that will be useful for pretraining models to address many different biological questions
the gLM is a novel way of representing genomic architecture and function simultaneously and could, as such be a very powerful resource

**Weaknesses:**

the gLM is poorly described and not validated, there are a few small vignettes describing it but these do not get at validation or generalizibility for the very interesting tasks described.  I have a hard time giving this paper the strongest recommendation because of these concerns.

**Questions:**

Is there a reason the gLM wasn't validated in some way? The examples seem very cherry picked, can you give us some sense of how you'd go about doing the validation for these very different biological questions (regulation, protein-protein interaction, etc). Do you think the gLM is really generalizable for all these different tasks? Where it struggles, why do you think it struggles?

---

> ### Author Response · Authors · 2024-11-21
> **Response to Reviewer HdWV**
>
> Thank you for your review and feedback. We appreciate that you find OMG to be a "very useful resource for the community" and that we "demonstrate its utility in training a novel mixed-modality genomic language model".
>
> > gLM is poorly described and not validated
>
> In section 4.3, we validate gLM2's performance on the DGEB benchmark (Section 4.3, Figure 3). As other reviewers mention, DGEB is not well described in the paper, and we have added the following Section 4.1: "We benchmark gLM2 on the Diverse Genomic Embedding Benchmark (DGEB) (West-Roberts, 2024).  DGEB is a comprehensive benchmark that evaluates model representations across diverse taxa and 18 tasks representing multiple axes of biological function, such as evolutionary distance similarity, remote homology prediction, enzyme classification, and retrieval sensitivity." Additionally, we added results on the ProteinGym benchmark in Appendix M.  We note that the ProteinGym benchmark includes eukaryotic sequences, which are poorly represented in the OMG dataset, and likely impacts gLM2 performance.
>
>
> > Can you give us some sense of how you'd go about doing the validation for  [protein-protein interaction]? Where it struggles, why do you think it struggles?
>
>
> We conducted the same PPI analysis on 32 evolutionarily conserved complexes with PDB structures previously characterized in [1]  and [2]. We include all results in Appendix P in Additional File 5. We find that for 16 complexes we detect at least 10 true positive co-evolutionary signals that match inter-chain contacts in PDB structure. However, not all co-evolutionary signals align well with physical contacts. We find that when there are distinct and localized patterns, gLM2 is more likely to capture such patterns, whereas, when these patterns are diffuse and contacts are spread out across the interface, gLM2 covariance metric is less accurate in detecting exact contacts. Additionally, previous studies [3, 4] have shown that covariance signal does not always align with PDB structures due to the dynamic nature of protein structure and protein-protein interactions, and the averaging effects categorical jacobian analysis results in for multiple stable conformations that may be present across distant taxons. Additionally, for structures that assemble in homo-multimers, PDB contacts do not correspond well with the combined covariance signal compressed in a single sequence. We refer to this additional analysis in section 4.4.
>
> > Can you give us some sense of how you'd go about doing the validation for  [regulation]? Where it struggles, why do you think it struggles?
>
> We have repeated the same regulatory element analysis in Figure 6 in E.coli K-12 MG1655 (U00096) genome, for which terminator and promoter annotations exist on EcoCyc. We identified 26 intergenic regions in the genome containing at least 1 promoter region and 1 promoter (sigma factor binding region) that is at least 500 base pairs long. We include all results in the zip file, where the terminator (green) and promoter (red) regions correspond with covariance signals directly. We include this zip file as Appendix P in Additional File 6. We demonstrate that the pattern observed in Figure 6 is recapitulated across the E.coli K-12 genome. We refer to this additional analysis in Section 4. 5.
>
>
> [1] https://openseq.org/cplx.php?sort=prob&order=DESC&mode=pdb
>
> [2] https://pmc.ncbi.nlm.nih.gov/articles/PMC4034769
>
> [3] https://www.pnas.org/doi/10.1073/pnas.2406285121
>
> [4] https://elifesciences.org/articles/02030)

---

### Official Review · Reviewer_jngi · 2024-11-04

**Soundness:** 2
**Presentation:** 2
**Contribution:** 2
**Rating:** 5
**Confidence:** 3

**Summary:**

The authors present open metagenomic datasets combined from two separate sources and filtered to maintain high-quality data. They further present the data set's utility in a trained model.

**Strengths:**

The authors give an excellent introduction to the field of metagenomics. They adequately describe the difficulties of balancing and filtering biological data sets and how some issues are exacerbated for metagenomic specifically. They detail each of the steps required to filter the data. Finally they open the data sets to the public to use.

**Weaknesses:**

While the methods for data filtering are explained, their impact is not always clear.
How many elements are removed for the edge-element removal?
For example, in the assembly filtering, a histogram of the number of invalid characters, why a threshold of 20% was chosen? What is the meaning of "(>3 CDS >7 ..."?
For the element length-based filtering section, again, what is the impact of why the specific threshold was chosen? Could you provide a supporting figure?
Essentially, This presents a novel data set whose importance is still unclear.

**Questions:**

The importance of the results is not explained; could you present this with other protein-to-protein-related works?
Which conclusion can I reach with this data that I couldn't other wise?

---

> ### Author Response · Authors · 2024-11-21
> **Response to Reviewer jngi**
>
> Thank you for taking the time to review our paper. We are glad that you find our paper provides background on the importance of releasing publicly-available metagenomic data for pretraining, as well as the challenges of balancing and filtering large-scale, unstructured metagenomic datasets.
>
> We appreciate your questions regarding data filtering, and we have updated our manuscript to clarify filtering steps, and provide supporting figures for specific thresholds used.
> >How many elements are removed for the edge-element removal?
>
> Thank you for this question. We have updated Section 3.1 with this information: "This filtering step removes ~1.4B genomic elements likely to be poor quality, partial sequences with high likelihood of assembly errors."
>
> > In the assembly filtering, a histogram of the number of invalid characters, why a threshold of 20% was chosen?
>
>  Thank you for raising this point. This filter is designed to remove genomic elements with poor assembly. To clarify the impact, we add a histogram to Appendix D, which shows distribution of invalid characters prior to filtering. Appendix D includes the impact of filtering: "The assembly quality filter removes elements containing more than 20% invalid characters, resulting in 0.004% of CDS and 0.2% of IGS being filtered from OMG"
>
> > For the element length-based filtering section, what is the impact of why the specific threshold was chosen?
>
> This filter is designed to exclude regions with poor gene-calls. To clarify the impact, we added to Section 3.1 " This filtering step removes 2.5e-5% of CDS , and 1e-4% of IGS elements from OMG". Appendix C shows the length distributions of CDS and IGS after filtering, and because only a small fraction of long tail data is removed, the overall data distribution is not affected. The threshold of 4000 bp for IGS sequences was determined to be able to include bonafide long IGS elements (e.g. 23S rRNAs that are ~3000bp long in microbial genomes). We used the threshold of 15,000 AAs for CDS sequences because there is a drop off in the frequency of proteins at 15,000 AAs in microbial genomes as previously identified in West-Roberts et al, 2023 [1] Fig 1A .
>
> > What is the meaning of "(>3 CDS >7...?
>
> This is referring to the filtering step in Section 3.1 "Contig length-based filtering and preprocessing", where we remove contigs lacking multiple genes. For N/X filtering, because we discard the invalid element and start a new contig, this may result in contigs which do not meet the length filter. In this case the contig is discarded.
>
>
> > The importance of the results is not explained.
>
> Our results from training gLM2 on the OMG dataset highlight the importance of **(1)** OMG's multi-gene context (unlike protein only datasets like UniRef50 used by ESM2) and **(2)**  OMG's scale and diversity (currently the largest and most diverse publicly available dataset for pretraining).
>
>
> **(1)** Multi-gene context allows zero-shot gLM2 prediction of protein-protein interactions. Understanding protein-protein interactions is critical for biological systems, because it defines molecular function. Existing sequence-based PPI prediction tools require supervision (e.g. PIPR [2], D-SCRIPT [3], Topsy-Turvy [4]), meaning they do not perform well for novel types of PPIs; this is a significant gap in the field because PPI data are extremely sparse. Furthermore, these tools cannot resolve interacting residues between proteins, an important aspect of PPI prediction task. Structure-based tools (e.g. Alphafold-multimer) typically assume that input proteins interact and also rely on supervision of known PDB interfaces. Therefore these tools have lower utility in predicting the presence of PPI or discovery novel PPIs in less characterized protein complexes. In our paper, we show gLM2 is capable of detecting covariance signal corresponding to PPI interfaces, a capability that existing foundation models lack.
>
>
> **(2)** OMG's scale and diversity is shown by clustering at 50% sequence identity, resulting in >3x more clusters than UniRef50 (OMG_prot50 described in Section 3), as well as gLM2's strong performance on DGEB (Figure 3). We believe that OMG, the largest and most diverse publicly available genomic dataset to date, will be a valuable resource to the community.
>
> [1] https://www.biorxiv.org/content/10.1101/2023.11.21.568195v1
>
> [2] https://academic.oup.com/bioinformatics/article/35/14/i305/5529260
>
> [3] https://www.cell.com/cell-systems/fulltext/S2405-4712(21)00333-1
>
> [4] https://academic.oup.com/bioinformatics/article/38/Supplement_1/i264/6617505

---

> > ### Comment · Reviewer_jngi · 2024-11-26
> >
> > Thank you for reviewing my comments I decided to increase the score.

---

### Meta-Review · Area_Chair_oomC · 2024-12-21

**Metareview:**

This paper introduces an open and large-scale metagenomics data called the "OMG dataset" and trains a mixed-modality genome language model called to gLM2 based on the constructed dataset.
Reviewers commend the authors' effort in constructing a large-scale and high-quality metagenomics dataset, as it may bring forth advances in the field by enabling the training advanced GLM for metagenomic studies in the future.
However, the gLM2 proposed by the authors and trained on the OMG dataset do not appear to be fully developed nor comprehensively evaluated and compared to other alternatives to demonstrate its potential advantages in a convincing manner.
Additional use cases of the constructed dataset for various predictive/analysis tasks in metagenomics and more comprehensive evaluations and comparisons of the resulting model(s) would be needed to demonstrate the potential benefits.

**Additional Comments On Reviewer Discussion:**

The authors have provided additional explanations and clarifications to address the reviewers' suggestions and concerns.
While gLM2 presented in this work doesn't yet seem to be fully developed/optimized to show clear merits and although the OMG dataset's usefulness needs to be further assessed, the AC believes that it would be beneficial to make the dataset available to the research community, enable researchers to further improve language modeling for metagenomics studies, and let the community assess the overall utility of the dataset and identify potential areas of improvement.

---

### Decision · Program_Chairs · 2025-01-22

Accept (Poster)